# Enhancing Lung Recellularization with Mesenchymal Stem Cells via Photobiomodulation Therapy: Insights into Cytokine Modulation and Sterilization

**DOI:** 10.3390/ijms251810131

**Published:** 2024-09-20

**Authors:** Leticia L. Guimarães, Auriléia A. Brito, Andressa D. Cereta, Ana Paula L. Oliveira, João Pedro R. Afonso, Diego A. C. P. G. Mello, Iransé Oliveira-Silva, Carlos H. M. Silva, Rodrigo F. Oliveira, Deise A. A. P. Oliveira, Rodolfo de Paula Vieira, Dante Brasil Santos, Giuseppe Insalaco, Luís V. F. Oliveira, Renata Kelly da Palma

**Affiliations:** 1School of Veterinary Medicine and Animal Sciences, University of São Paulo, São Paulo 05508-220, SP, Brazil; leleticialopesg@usp.br (L.L.G.); adcereta@hotmail.com (A.D.C.); rekellyp@hotmail.com (R.K.d.P.); 2Post-Graduate Program in Biophotonics Applied to Health Sciences, Nove de Julho University (UNINOVE), São Paulo 17011-102, SP, Brazil; leiaapbritto@gmail.com (A.A.B.); apligeiro@gmail.com (A.P.L.O.); 3Human Movement and Rehabilitation, Post-Graduate Program, Evangelic University of Goiás, UniEVANGELICA, Anápolis 75083-515, GO, Brazil; joaopedro180599@gmail.com (J.P.R.A.); diego0611escs@hotmail.com (D.A.C.P.G.M.); iranseoliveira@hotmail.com (I.O.-S.); carloshmendes@unievangelica.edu.br (C.H.M.S.); rodrigofranco65@gmail.com (R.F.O.); deisepyres@gmail.com (D.A.A.P.O.); rodrelena@yahoo.com.br (R.d.P.V.); dantebsantos@gmail.com (D.B.S.); 4Institute of Translational Pharmacology, National Research Council of Italy (CNR), 90146 Palermo, Italy; giuseppe.insalaco@ift.cnr.it; 5Facultad de Ciencias de la Salud de Manresa, Universitat de Vic-Universitat Central de Catalunya (UVic-UCC), 08500 Manresa, Spain

**Keywords:** photobiomodulation therapy, extracellular matrix, lung, decellularization

## Abstract

Several lung diseases can cause structural damage, making lung transplantation the only therapeutic option for advanced disease stages. However, the transplantation success rate remains limited. Lung bioengineering using the natural extracellular matrix (ECM) of decellularized lungs is a potential alternative. The use of undifferentiated cells to seed the ECM is practical; however, sterilizing the organ for recellularization is challenging. Photobiomodulation therapy (PBMT) may offer a solution, in which the wavelength is crucial for tissue penetration. This study aimed to explore the potential of optimizing lung recellularization with mesenchymal stem cells using PBMT (660 nm) after sterilization with PBMT (880 nm). The lungs from C57BL/6 mice were decellularized using 1% SDS and sterilized using PBMT (880 nm, 100 mW, 30 s). Recellularization was performed in two groups: (1) recellularized lung and (2) recellularized lung + 660 nm PBMT (660 nm, 100 mW, 30 s). Both were seeded with mesenchymal stem cells from human tooth pulp (DPSc) and incubated for 24 h at 37 °C and 5% CO_2_ in bioreactor-like conditions with continuous positive airway pressure (CPAP) at 20 cmH_2_O and 90% O_2_. The culture medium was analyzed after 24 h. H&E, immunostaining, SEM, and ELISA assays were performed. Viable biological scaffolds were produced, which were free of cell DNA and preserved the glycosaminoglycans; collagens I, III, and IV; fibronectin; laminin; elastin; and the lung structure (SEM). The IL-6 and IL-8 levels were stable during the 24 h culture, but the IFN-γ levels showed significant differences in the recellularized lung and recellularized lung + 660 nm PBMT groups. Greater immunological modulation was observed in the recellularized groups regarding pro-inflammatory cytokines (IL-6, IFN-γ, and IL-8). These findings suggest that PBMT plays a role in cytokine regulation and antimicrobial activity, thus offering promise for enhanced therapeutic strategies in lung bioengineering.

## 1. Introduction

Lung diseases such as chronic obstructive pulmonary disease (COPD), idiopathic pulmonary fibrosis, primary pulmonary arterial hypertension, interstitial pulmonary disease, cystic fibrosis, and α-1-antitrypsin deficiency result in structural lung damage changes, with lung transplantation being a therapeutic indication when the disease reaches advanced progression [1]. Unfortunately, the success of lung transplantation is limited, mainly due to the scarcity of organ donors and the incidence of bronchiolitis obliterans, which results from an alloimmune response caused by disparities between the donor and the recipient’s antigens. The 50% survival rate after lung transplantation is approximately five years [2]. Lung bioengineering has emerged as a promising alternative to address these challenges.

This field has achieved significant progress by leveraging advances in decellularization techniques and scaffold development to create platforms for cell seeding and tissue regeneration [3,4]. Decellularized lung extracellular matrix (ECM) scaffolds provide a biomimetic environment that supports cell attachment, proliferation, and differentiation [5,6,7,8]. The use of undifferentiated cells to seed an ECM may be a more practical strategy for lung bioengineering because of the possibility of expanding these cells and their ability to differentiate into different phenotypes. However, this approach requires stem cells to be differentiated into the necessary phenotypes at specific sites within the organ structure [6,7]. Patient-derived mesenchymal stem cells (MSCs) have been demonstrated as an ideal cell type owing to their self-renewal ability and ability to differentiate into various cell types. Furthermore, owing to their feasibility for isolation and embryonic origin, dental pulp mesenchymal stem cells (DPSCs) are of great interest for use in regenerative medicine [9].

Scaffolds derived from human lungs would be the most appropriate for clinical applications owing to their compatibility and structural integrity. Although mouse ECMs are suitable for preclinical studies, translating these findings into human applications requires the use of decellularized human lungs or potentially porcine lungs, which are similar in size and structure to human lungs [10]. Recent studies have highlighted the transition from small animal models, such as mice, to larger animal models and whole human lungs for preclinical testing and therapeutic development in lung bioengineering [11,12]. This underscores the necessity of utilizing clinically relevant models to bridge the gap between laboratory research and human applications [13].

In addition to the cell type, the sterilization of the organ presents another challenge during lung recellularization. Previous studies have shown that the methodologies applied for ECM sterilization, such as gamma irradiation, can alter the mechanical properties of the tissue, thus impairing the ECM–cell interactions [11]. Therefore, a new tool and methodology for sterilization without changing the mechanical properties of decellularized lungs is necessary. Photobiomodulation therapy (PBMT) has emerged as a potential alternative treatment, and previous studies have shown its capacity to modulate cell growth, differentiation, proliferation, self-renewal, and survival [12,13].

In addition to being a potent alternative for sterilization, PBMT has been demonstrated to be effective in differentiating stem cells and plays an anti-inflammatory role in pulmonary inflammation [14,15]. However, it is well known that the wavelength is one of the key factors in PBMT that regulates the depth and penetration of laser irradiance into the tissue [16]. Therefore, we chose an 808 nm wavelength (in the near-infrared spectral region) for sterilization and a 660 nm wavelength (in the red light spectral region) for recellularization. The objective of this study was to verify whether PBMT (660 nm) could optimize lung recellularization with mesenchymal stem cells by cytokine modulation after sterilization by PBMT (880 nm).

## 2. Results

### 2.1. Decellularization Assessment

Lung decellularization was successfully performed according to the protocol described in the Methods section, which lasted approximately 3–4 h. Through this process, it was possible to verify changes in the organ color, which changed from reddish to translucent (Figure 1A). The lungs were monitored throughout the decellularization process while ensuring constant pressure of 20 cmH_2_O for the reagents, as well as tracheal (transpulmonary) pressure of 20 cmH_2_O. It is worth mentioning that decellularization was performed on the entire organ, which could preserve the trachea that was used in connection to the constant ventilatory positive pressure during the recellularization process.

Lung scaffolds obtained by decellularization showed genomic DNA content in the range of 17.75 ± 2.05 ng/mg (below the 50 ng/mg suggested by Crapo et al. [17]) (Figure 1B) and lacked cellular nuclei, as assessed using hematoxylin and eosin (H&E) and Colloidal iron staining (Figure 2).

We performed immunostaining to analyze the presence of collagens I, III, and IV and fibronectin, laminin, and elastin in the recellularized lung. The results showed that the ECM structure was maintained after recellularization and that the main matrix proteins were preserved. Immunoreactivity was evident for all analyzed proteins, indicating that PBMT did not negatively affect the ECM structure. Figure 3 illustrates these observations, highlighting the integrity of collagens I, III, and IV and fibronectin, laminin, and elastin. Scanning electron microscopy (SEM) analysis showed that the microscopic structure of the lungs was maintained after recellularization. We observed the preservation of collagen and elastin fibers, as well as the integrity of the alveoli and vascular structures. Figure 4 shows representative SEM images, in which the intact alveolar architecture and the uniform distribution of collagen fibers can be seen (Figure 4).

### 2.2. Lung Scaffold Recellularization

The quantification of the IL-6, IL-8, IFN-γ, and IL-10 cytokine levels was performed to evaluate the inflammatory response during the recellularization process (Figure 5). IL-6 and IL-8 are pro-inflammatory cytokines and their stable levels during the procedure indicate the absence of significant inflammation. IFN-γ, also a pro-inflammatory cytokine, showed significant differences between the recellularized groups, thus suggesting immune modulation by PBMT. In contrast, IL-10, an anti-inflammatory cytokine, was reduced after PBMT, which can be attributed to the absence of an initial inflammatory stimulus in the recellularized cells. These results highlight the modulatory role of PBMT in the immune response during lung recellularization.

## 3. Discussion

In this study, we investigated the potential of PBMT in lung bioengineering, focusing on its applications in ECM sterilization and cytokine modulation. Our findings suggest that PBMT may be a valuable tool in this field. Here, we show three potential benefits: (i) PBMT can be used as a tool for acellular ECM sterilization, (ii) PBMT increases cell proliferation, and (iii) PBMT and the ECM play a role in cytokine modulation.

The production of biomaterials is a promising technique, achieved through different approaches in the process of decellularization, which makes it possible to obtain a biological scaffold with the ability to maintain adequate physiological function to consequently perform a transplant [8]. Considering this, we were able to obtain a biological scaffold through the decellularization protocol carried out through the pulmonary artery, owing to the efficiency of the process, which can be completed within an average of 3 to 4 h. In addition, this protocol allows the control of the perfusion pressure in the pulmonary artery (physiological pressure of 20 cmH_2_O), which may prevent barotrauma [18].

The lungs were effectively decellularized (Figure 2), resulting in biological scaffolds with cell content removed and an extracellular matrix that was preserved and translucent by perfusion with the chemical agent SDS, an ionic detergent. We chose SDS because this ionic detergent can remove cells without inducing significant changes in the micromechanical properties of acellular lungs [19]. In addition to cell nuclei removal, visualized by H&E and demonstrated by DNA quantification, the preservation of the ECM is relevant for organ bioengineering. Our protocol was efficient in maintaining ECM proteins, as demonstrated by the presence of collagens I, III, and IV; fibronectin; laminin; and elastin (Figure 4). The glycosaminoglycans and the intact alveolar and vascular structures of the lungs were also shown (Figure 3 and Figure 5). These results satisfy all of the criteria for successful lung decellularization [17].

Recent studies have shown the potential benefits of using MSCs, which can be isolated from patients from sources such as tooth pulp and expanded in culture to physiologically relevant numbers. In addition to their fibroblastic morphology, self-renewal, and full ability to differentiate into various cell types, MSCs have demonstrated efficient adherence to lung structures regardless of the decellularization method, sterilization method, storage time, or disease state. Thus, MSCs have a wide range of applications in recellularization and are a physiologically relevant cell source for this process [11,20,21]. Although we propose that PBMT may increase cell proliferation, our study did not include quantitative data to support this claim. Future studies should involve direct comparisons of the cell growth rates between PBMT-treated and untreated groups to validate this potential benefit.

Apart from the cell source, PBMT has shown excellent results and is considered an alternative approach for recellularization. PBMT can modulate cell growth, viability, and differentiation and induce increased cell proliferation, self-renewal, survival, and angiogenesis [15,22]. However, an important issue that needs to be considered before recellularization is scaffold sterilization, which eliminates any risk of transmission of viruses and bacteria from the tissue/organ donor to the transplanted receptor. Different sterilization methods have been used; however, they usually involve aggressive sterilization, which deteriorates the structural components and alters the mechanical performance of tissue [11,23,24]. Our work proposes a new sterilization methodology where PBMT can not only assist in cell proliferation through irradiation at an infrared light wavelength (660 nm), but can also assist in sterilization by means of irradiation at a red light wavelength (808 nm). Future studies should include experiments to quantify the bacterial load reduction, such as culturing on blood agar, colony counting, or PCR for the 16s rRNA of common microorganisms, to confirm the efficacy of PBMT in eliminating bacterial contamination.

PBMT has been described as a tool with the potential to decrease disease severity through the positive regulation of IL-10 release. Previous studies associated with PBMT and mesenchymal stem cells in an experimental model of COPD showed increased expression and levels of IL-10 in the lungs [25,26]. Furthermore, PBMT modulated the immunological response of bleomycin-stimulated lung fibroblasts, leading to increased anti-inflammatory IL-10 secretion. However, in this study, we expected a greater immunological response from the acellular ECM considering the biological signs that were maintained. In fact, greater immunological modulation was observed in the recellularized groups (recellularized and recellularized + PBMT) regarding pro-inflammatory cytokines (IL-6, IFN-γ, and IL-8). In contrast, PBMT decreased IL-10 release, differing from studies related to diseases, most likely because cells do not need to be activated by inflammatory processes. However, further detailed analyses and experiments are necessary to clearly delineate these effects and their underlying mechanisms.

This study provides preliminary insights for future research in the field of lung bioengineering. By addressing these limitations and building on these findings, subsequent studies could further explore the therapeutic potential of PBMT and the ECM in lung regeneration.

## 4. Materials and Methods

The experimental procedures were approved by the Ethics Committee for Animal Research of the University of São Paulo and carried out in accordance with the National Institutes of Health Guide for the Care and Use of Laboratory Animals (NIH Publications No. 8023, revised 1978).

### 4.1. Lung Decellularization

Lungs were obtained from 15 male mice C57/BL6, weighing 17 to 20 g. Initially, the mice were anesthetized with xylazine and ketamine (1 mg/kg, intraperitoneal) according to the standard protocol adopted by the laboratory and subsequently sacrificed by exsanguination through the abdominal aorta [22]. Immediately after euthanasia, the diaphragm was punctured and the rib cage was cut to reveal the lungs. The pulmonary artery was cannulated, and the mice’s lungs were perfused through the right ventricle with a phosphate buffer solution (PBS) containing 50 U/mL heparin and 1μg/mL sodium nitroprusside (Sigma Chemical Co., St. Louis, MO, USA) to prevent the formation of blood clots. Finally, the heart, lungs, and trachea were dissected and removed en bloc and stored in a −80 °C freezer until the decellularization process was performed.

The lung decellularization protocol included collection, cleaning, freezing, thawing, and washing with 1% sodium dodecyl sulfate (SDS) and PBS, as previously described [18,27]. Thirteen lungs were placed in the experimental system after the cannulation of the trachea and the pulmonary arteries. The trachea was cannulated and connected to a continuous positive airway pressure (CPAP) device that was set to provide tracheal (i.e., transpulmonary) pressure of 20 cmH_2_O to inflate the lung to a physiological volume to prevent atelectasis. The following decellularization steps were performed through the pulmonary artery: (1) PBS 1× for 30 min, (2) deionized water for 15 min, (3) 1% SDS for 150 min, and (4) PBS for 30 min at pressure of 20 cmH_2_O.

### 4.2. Histological and Immunohistochemistry Analysis of Samples

After decellularization, the lung scaffold (as well as the native lung for comparison) was divided into lobes, fixed in paraformaldehyde at 4% for 48 h and then dehydrated, diaphanized, and embedded in paraffin. Subsequently, 5 µm sections were created with the aid of a microtome (RM2265, Leica—Nussloch, Germany), and hematoxylin and eosin (HE) and colloidal iron stains were used to locate possible remaining cell nuclei in the ECM. The counting of the central nuclei was performed using the ImageJ program (version 1.39u), using the “cell counter” plugin.

For immunohistochemistry, the sections were rehydrated and microwaved (1 min at 160 W) in citrate buffer (1.83 mM monohydrate citric acid and 8.9 mM sodium citrate tribasic dehydrate; pH 6.0) for antigen retrieval. An endogenous peroxidase block was performed using 3% hydrogen peroxide in distilled water for 30 min in the dark. Nonspecific protein interactions were blocked with 2% bovine serum albumin (BSA) in PBS for 30 min. Then, the slides were incubated with primary antibodies, namely anti-collagen I (# 600-401- 103S, 1:100, Rockland—Limerick, ME, USA), anti-collagen III (# sc-8779, 1:100, Santa Cruz Biotechnology—Dallas, TX, USA), anti-collagen IV (# 1-CO083-0, 1:100, Quartett—Berlin, Germany), anti-laminin alfa-2 subunit (# bs-8561R, 1:100, Bioss Antibodies—Woburn, MA, USA), and anti-fibronectin (# NBP1-91258, 1:200, Novus Biologicals—Littleton, CO, USA), overnight in a humid chamber at 4 °C. For negative controls, irrelevant anti-mouse IgG (# M5284, Sigma) or anti-rabbit IgG (# ab27478, Abcam—Cambridge, UK) were used under the same conditions to replace the primary antibody. The reaction was detected using a Dako Advance HRP kit that included a secondary antibody (# K6068, Dako—Carpinteria, CA, USA), and the color was developed using DAB (# K3468, Dako). The slides were slightly counterstained with hematoxylin. Between each step of antibody incubation, the slides were washed with PBS containing 0.2% BSA. Finally, the slides were assembled and visualized under a Nikon Eclipse 80I microscope (Boston, MA, USA).

### 4.3. Scanning Electron Microscopy (SEM)

Samples from the control and decellularized lungs (one lobe each) were prepared for analysis using a scanning electron microscope, following a standard protocol for the preparation of tissue samples. The samples were fixed with 2% glutaraldehyde and 2.5% paraformaldehyde in 0.1 M cacodylate buffer (EMD Biosciences, Darmstadt, Germany) for 2 h at room temperature; they were then washed with distilled water in a computerized ultrasonic washer (UltraSonicCleaner, Darmstadt, Germany) and dehydrated in different concentrations of 70% alcohol for 24 h, followed by 80% alcohol, 90%, and 100% for 10 min each; then, dry dehydration was performed at a critical point (Balzers CPD 020). Subsequently, the material was placed on a metallic support for gold plating (“sputtering” Emitech K550, Darmstadt, Germany). We used a scanning electron microscope (Hitachi Analytical Table Top Microscope TM3000, Hitachi, Darmstadt, Germany) at 15 kVa acceleration to observe the results.

### 4.4. Quantification of DNA in Decellularized Lungs

Five-lobe lung scaffolds (as well as five native lungs for comparison) were fixed via the bronchial infusion of a 3:1 ratio mixture of the Optimal Cutting Temperature compound (OCT, Sakura, Japan) and PBS. Cryosections (12 mm) of frozen lung samples were obtained using a cryostat (HM 560 CryoStar; Thermo Scientific, Waltham, MA, USA). DAPI staining was used to verify the absence of DNA nuclei after decellularization. For DAPI staining, a stock solution of DAPI (5 mg/mL; 14.3 mm for dihydrochloride or 10.9 mm for dilate) was prepared by dissolving 10 mg of DAPI in 2 mL of deionized water, followed by sonication for 2 h.

The amount of DNA remaining in the decellularized samples was extracted as previously described [18]. A sample from the right middle lobe of each lung was dried and weighed, and the total genomic DNA was isolated using a spin-column-based PureLinks Genomic DNA Mini Kit (Invitrogen™, Waltham, MA, USA), according to the manufacturer’s instructions. The yield of double-stranded DNA was measured using spectrophotometry (NanoDrop 1000, Thermo Scientific) and normalized to the sample tissue weight.

### 4.5. Isolation and Culture of Dental Pulp Stem Cells (DPSCs)

Stem cells derived from dental pulp were obtained from healthy patients, as provided by Prof. Dra. Auriléia Aparecida de Brito. After tooth extraction, they were kept in a container with PBS 1× and a storage solution consisting of 3 mL αMEM (Minimum Essential Medium Eagle—α modification, Sigma^®^, M4526) with 2% antibiotic–antimycotic solution (Gibco^®^, A5955, Darmstadt, Germany), 10% fetal bovine serum (Fetal Bovine Serum, Gibco^®^, 10437028), and 2 mM L-glutamine (Gluta MAXTM, Gibco^®^, 35050061) [15,28,29]. Immediately after the extraction of the dental pulp, the MSCs were separated and purified by gradient centrifugation and transferred to a culture plate with medium consisting of αMEM (Sigma^®^, M4526) supplemented with 1% antibiotic–antimycotic solution (Gibco^®^, A5955), 10% fetal bovine serum (FetalClone^®^ III, Hyclone^®^, SH 30109.03), 100 μM ascorbic acid, and 2 mM L-glutamine (GlutaMAXTM, Gibco ^®^, 35050061) and kept in an oven at 37 °C, under 5% CO_2_ and 95% of atmospheric air [29]. Cell growth was observed daily under an inverted microscope to observe the morphological characteristics until they reached 80% confluence for subculture, and the culture medium was monitored and changed every 24 h for cell proliferation.

### 4.6. Characterization of Dental Pulp Stem Cells (DPSCs)

After the fourth passage, the cells were removed to analyze the degree of purity. The results confirmed the phenotypic characterization of these stem cells by flow cytometry, which revealed that the expression of CD105 (SH2) and CD73 (SH3/4) was negative for the markers of hematopoietic lineage CD34 (Figure 6A).

### 4.7. Cultivation of DPSCs in Decellularized Lungs

After decellularization, the DPSCs were isolated and expanded in culture. The cells were suspended in the culture medium and carefully seeded onto the decellularized mouse extracellular matrix (ECM). Lung recellularization was performed in 12 decellularized lungs for 24 h, and the lungs were divided into two groups. In the recellularized lung group (n = 6), the lungs were seeded with 1 × 106 dental pulp stem cells, and, in the recellularized lung + 660 nm PBMT group (n = 6), the lungs were seeded with 1 × 106 dental pulp stem cells and irradiated with PBMT at 880 nm.

Initially, the DPSCs (1 × 106) were suspended in 1 mL of RPMI 1640 medium supplemented with 15% SFB, 2 mM glutamine, and 1% of penicillin–streptomycin [18]; filtered through a 40 µm cell filter to remove groups of cells; and injected into the left and right bronchi of the decellularized lungs using a 20-gauge catheter. The lungs were placed with the medium in a 25 mL falcon tube of ideal size to simulate a rib cage and sterilized with PBMT by irradiation at 660 nm in the red light wavelength, at a dose of 3 joules (30 s each), before DPSC seeding (Figure 6B). To induce cell proliferation, the recellularized lung + PBMT group was irradiated with infrared light at 880 nm at the same dose of 3 joules (30 s each). The lungs from both groups were left in an incubator to allow the cultivation of cells at 37 °C under 5% CO_2_ and 95% of atmospheric air in pulmonary differentiation medium, entirely sealed and sterile, remaining for a culture period of 24 h [8] and maintaining a ventilatory method with constant positive pressure of 20 cmH_2_O and 90% O_2_ [30]. During the recellularization process, 2 mL of the medium was removed and the same amount of fresh medium was added every 12 and 24 h for ELISA evaluation.

For comparison with the recellularized groups, we performed a DPSC flask culture with and without PBMT irradiation under the same conditions. The main difference between the two-dimensional (2D) and three-dimensional (3D) cultures was demonstrated.

### 4.8. Evaluation of Cytokine Levels by ELISA

The levels of IL-1β, IL-6, IL-8, IFN-γ, and IL-10 were assessed using the BioLegend and R&D Systems kit. ELISA is short for enzyme-linked immunosorbent assay, which allows the detection of specific antibodies.

### 4.9. Statistical Analysis

Statistical analyses were performed using the GraphPad Prism 8 software. All values are expressed as the mean and standard error of the mean (SEM). Comparison values of the control and decellularized groups were obtained using a paired test group. ANOVA was used to compare the DPSc, DPSc + PBMT, recellularized lung, and recellularized lung + PBMT groups during the recellularization process, and Tukey’s post hoc test was used. Statistical significance was set at *p* < 0.05.

## 5. Conclusions

In conclusion, the results of this study suggest that low-level laser irradiation modulates cytokine levels and antimicrobial activity during lung recellularization with mesenchymal stem cells. Furthermore, the extracellular matrix (ECM) may play a significant role in determining the cytokine levels, which should be considered in future studies.

## Figures and Tables

**Figure 1 ijms-25-10131-f001:**
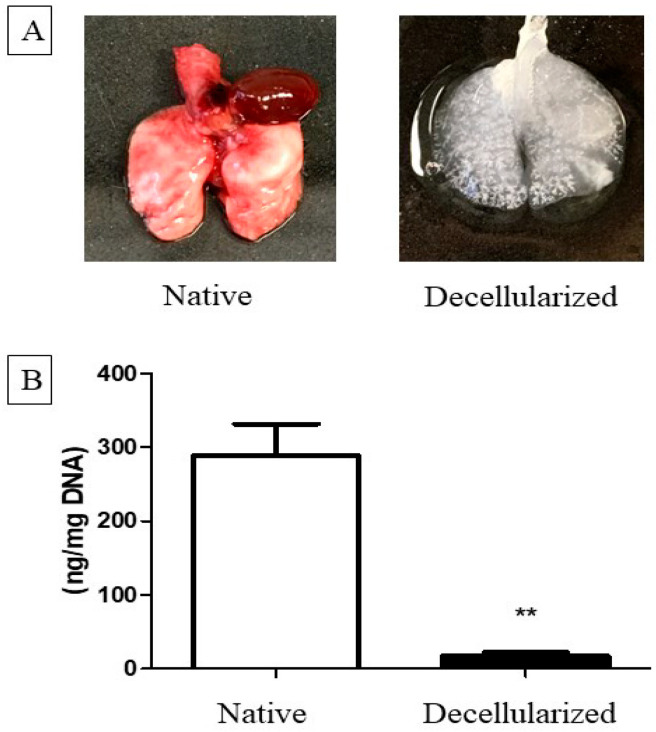
(**A**) Lungs before (native) and after decellularization process. (**B**) Analysis of DNA quantification in decellularized lungs (n = 5). Data represent mean ± standard error (SEM). The asterisk indicates the significance of the difference between the groups (** *p* < 0.001). Lung scaffolds obtained by decellularization showed genomic DNA content in the range of 17.75 ± 2.05 ng/mg (below the 50 ng/mg suggested by Crapo et al. [17] (**B**).

**Figure 2 ijms-25-10131-f002:**
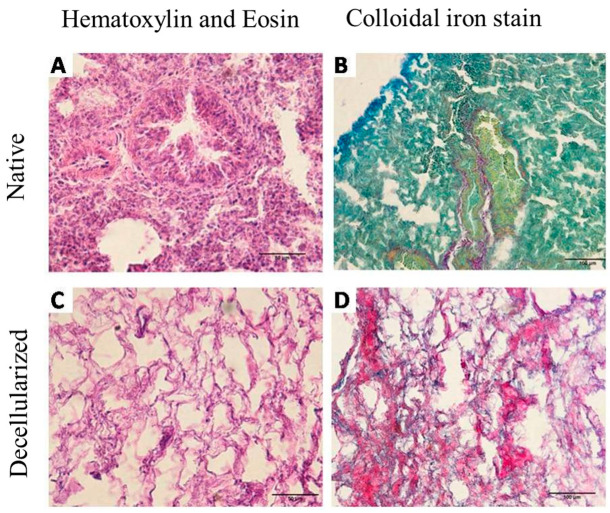
Representative native and decellularized lung tissue was visualized using hematoxylin and eosin (H&E) and colloidal iron staining. (**A**,**C**) H&E; (**B**,**D**) colloidal iron staining. The sections indicated a lack of visible nuclear material (**B**,**D**). Scale bar = 50 µm (**A**,**C**) and =100 µm (**B**,**D**).

**Figure 3 ijms-25-10131-f003:**
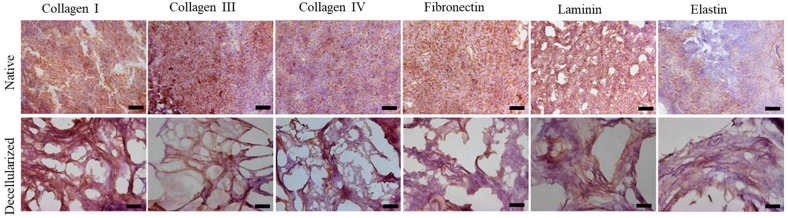
Immunohistochemical analysis (one lobe). Presence of collagens I, III, and IV and fibronectin, laminin, and elastin. Scale bar = 100 µm.

**Figure 4 ijms-25-10131-f004:**
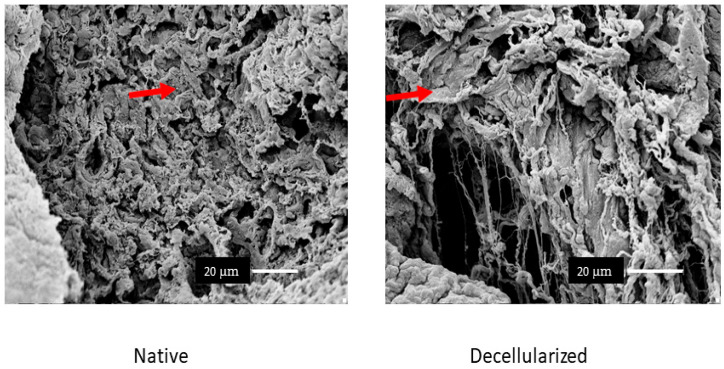
Scanning electron microscopy images. The sections indicate the maintenance of the tissue architecture indicated by the red arrows. Scale bar = 20 µm.

**Figure 5 ijms-25-10131-f005:**
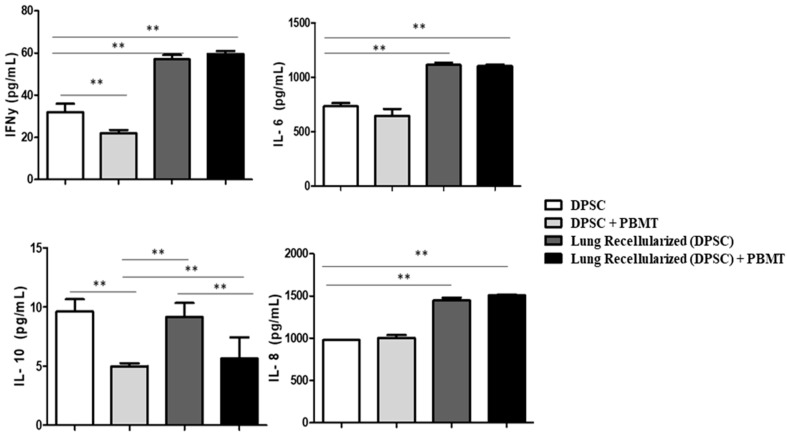
Effect of PBMT therapy on lung recellularization regarding the levels of IL-6, IL-8, IFN-y, and IL-10 after 24 h in the cell culture supernatant of tooth pulp stem cells. Data represent mean ± standard error (SEM). Asterisks indicate significant differences between the groups (** *p* < 0.001).

**Figure 6 ijms-25-10131-f006:**
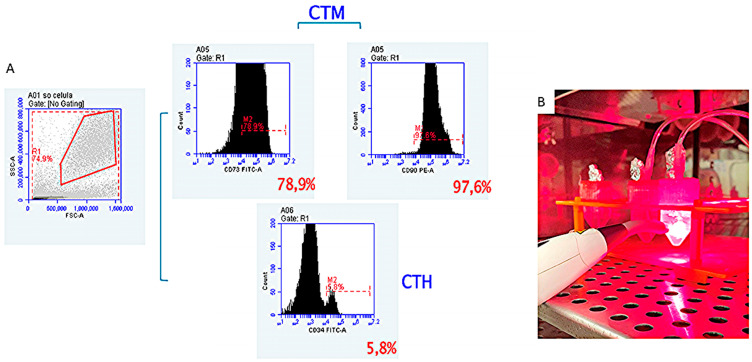
(**A**) Characterization of dental pulp stem cells (DPSCs). Passage 4 of the culture was used and the phenotypes of the cells showed that they were positive for the mesenchymal markers CD90 and CD73 (upper histograms), but did not express the hematopoietic markers, such as CD34 (lower histogram). (**B**) Sterilization and cell proliferation. Sterilization by irradiation with red light (660 nm) and cell proliferation by irradiation with infrared light (880 nm).

## Data Availability

The data generated in this study are available to the scientific community upon request.

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
