# Peer review of "Enhancing Lung Recellularization with Mesenchymal Stem Cells via Photobiomodulation Therapy: Insights into Cytokine Modulation and Sterilization"

_ijms, 2024, doi:10.3390/ijms251810131_

Round 1

Reviewer 1 Report

Comments and Suggestions for Authors

The authors present the use of a visible light irradiation technique to neutralize possible bacterial contamination and cellular stimulation to improve the repopulation of lung scaffolds. However, there are serious deficiencies in the presentation of the manuscript; the figure mentioned is missing, and others are mixed. There is a poor description of the results and an even poorer interpretation. Some experiments are missing; if they were done, they should be presented here.

The abstract is confusing and does not separate the ideas well; for example, the experimental groups are not identified. They mention LLL at 660 nm and then change it to 808 nm; it is not clear what the difference is. It is not clear what absent DNA content means; is it bacterial? Cellular? Talks about LLL and LLLT, please unify. In the objective of the research, should there not be cells decontaminated or free of microorganisms? It doesn't mention the role of cytokines, and we don't know if reducing IL-10 levels is good or bad. Please, let us clear about the use of the mouse scaffold and human pulp. The conclusions should begin with mentioning the antimicrobial activity, which also regulates components of the innate immune response.

In the introduction, they mention the convenience of regenerating lungs by bioengineering, by repopulation with undifferentiated mesenchymal cells is an alternative. I would like to be mentioned which scaffold you think the technique could be performed on for clinical applications. For preclinical studies, mouse ECMs are fine, but what would they be for humans?

In the results, the description of DNA detection is poor, and it does not mention that it is ng of DNA per mg of tissue. It refers to a not-explained comparison so that it is clear native = tissue without decellularization, for example. The figure needs to be improved. It looks vertically squashed.

In general, there is little concern for a job well done, for example;

·       On line 86, it must be Figure 1.

·       In lines 121 and 122, the figure numbers are confused.

·       Line 113 says FFigure 1.

·       Line 151 says removal of debris and blood…., but where?

·       It is not indicated what we should see in panels A, B, C, and D of Figure 2,

·       It takes little advantage of Figure 2; it does not describe what a normal lung with its cellularity should look like in contrast to a decellularized one.

·       Line 158: I think you are still talking about figure 3

·       Line 160 it must be Figure 4

·       Line 165, since a figure is missing, check the figure numbers.

It also does not take advantage of immunostaining; it does not describe what is seen, and immunoreactivity is evident.

Online 158-160 reports the results of the SEM, indicating that the structure of the lungs is maintained but does not indicate how this maintenance of structure is evident.

Lines 161-163 talk about the notorious improvement in cell growth…. which cells, the ECM, the ECM with pulp cells, only the pulp cells; finaly it refers to a figure that does not exist.

There is no description of how pulp cells are mixed with mouse ECM or how they were treated with LLLT and its controls.

We can presume those experiments were done, as shown in the legends in Figure 5.

Line 164-168, refers to cytokine determinations, but neither here nor in the introduction do they indicate how this parameter should be interpreted, for example, that IL-6 and IL-8 are proinflammatory cytokines and that their stable levels during the procedure, then remain in the absence of inflammation. What happens with IFN-gamma and IL-10?

Discussion.

Lines 210 -213 should be removed.

Line 214, begins the discussion by making a value judgment of his own work, calling it “remarkable”. In lines 215 -217 they lists the notable advances of this work.

i) ensure that, “LLLT can be used as a tool 215 for acellular ECM sterilization”. However they do not have any experiments that show a decrease or absence of bacterial load, such as culture on blood agar, colony count, PCR for 16s of the most common microorganisms. Or an experiment were contaminate tissues with a certain amount of bacteria are treated and verified if the procedure eliminates them with its controls.

ii) they assure that, “LLLT increase cell proliferation”, however they do not show any results where we can see an increase in the speed of cell growth comparing groups treated, not treated with ECM without LLLT etc....

iii) they claim that “LLLT and ECM play 216 a role in cytokine modulation”, but it is not clear where they see this assertion.

The only remarkable thing is the lack of self-criticism and rigor in the preparation of this manuscript and the lack of concern about the work being carried out consistently.

On line 217—218, in the first place, this work should have to be published for “this work will be relevant for future studies in the field of lung bioengineering”, we will see! 

In materials and methods

It is better written, but the parameters also change; for example, they indicate a CPAP of 10 cm H2O and then 20 cm H2O without mentioning that they are different sites. 

I didn't want to continue reading, so the authors would have to pass the manuscript on to someone with more experience gathering the information, interpreting the results, and planning the experiments.

Comments on the Quality of English Language

Can be improved

Author Response

Dear Reviewers (To Reviewer 1)

We want to extend our sincere gratitude for your thorough and insightful comments on our manuscript. Your feedback has been invaluable in refining our work, and we believe that the manuscript has significantly improved due to your constructive critiques. We are pleased to resubmit our revised manuscript titled: "Enhancing Lung Recellularization with Mesenchymal Stem Cells via Photobiomodulation Therapy: Insights into Cytokine Modulation and Sterilization."

In response to your comments, we have made substantial revisions to address your concerns and improve the clarity and robustness of our study. Below, we provide detailed responses to each of the points raised, outlining the changes made and the rationale behind our modifications.

We are confident that the revised manuscript now meets the high standards required for publication and provides valuable insights into lung recellularization.

Thank you once again for your critical evaluation and support.

Best regards,

Renata Kelly da Palma

Reviewer 1

The authors present a visible light irradiation technique to neutralize possible bacterial contamination and cellular stimulation to improve the repopulation of lung scaffolds. However, there are serious deficiencies in the presentation of the manuscript; the figure mentioned is missing, and others are mixed. The results are poorly described and interpreted. Some experiments are missing; if they were done, they should be presented here.

The abstract is confusing and does not separate the ideas well; for example, the experimental groups are not identified. They mention LLL at 660 nm and then change it to 808 nm; it is not clear what the difference is.

Thank you for your comment. We included the information.

It is not clear what absent DNA content means; is it bacterial? Cellular?

We included the information.

Talks about LLL and LLLT, please unify.

We unified.

In the objective of the research, should there not be cells decontaminated or free of microorganisms?

Actually, we sterilized a scaffold that no longer contained any cells.

It doesn't mention the role of cytokines, and we don't know if reducing IL-10 levels is good or bad.

We include this information

Please, let us clear about the use of the mouse scaffold and human pulp.

We include this information

The conclusions should begin with mentioning the antimicrobial activity, which also regulates components of the innate immune response.

We include this information

In the introduction, they mention the convenience of regenerating lungs by bioengineering, by repopulation with undifferentiated mesenchymal cells is an alternative. I would like to be mentioned which scaffold you think the technique could be performed on for clinical applications. For preclinical studies, mouse ECMs are fine, but what would they be for humans?

We include this information

In the results, the description of DNA detection is poor, and it does not mention that it is ng of DNA per mg of tissue.

We include this information.

It refers to a not-explained comparison so that it is clear native = tissue without decellularization, for example. The figure needs to be improved. It looks vertically squashed.

We improved it.

In general, there is little concern for a job well done; for example;

  • On line 86, it must be Figure 1.

        We correct it.

  • In lines 121 and 122, the figure numbers are confused.

       We correct it.

  •  

   Line 113 says FFigure 1.

       We correct it.

  • Line 151 says removal of debris and blood…., but where?

       We correct it.

  • It is not indicated what we should see in panels A, B, C, and D of Figure 2

       We include it.

  • It takes little advantage of Figure 2; it does not describe what a normal lung with its cellularity should look like in contrast to a decellularized one.

       We include this information.

  • Line 158: I think you are still talking about figure 3

      We correct it.

  • Line 160 it must be Figure 4

            We correct it.

  • Line 165, since a figure is missing, check the figure numbers.

          We correct it.

It also does not take advantage of immunostaining; it does not describe what is seen, and immunoreactivity is evident.

We include this information

Online 158-160 reports the results of the SEM, indicating that the structure of the lungs is maintained but does not indicate how this maintenance of structure is evident.

We include this information

Lines 161-163 talk about the notorious improvement in cell growth…. which cells, the ECM, the ECM with pulp cells, only the pulp cells; finaly it refers to a figure that does not exist.

We rewrote this text, it was actually a translation error. Cell growth was carried out before seeded in ECM, just to achieve viability for this seeding.

There is no description of how pulp cells are mixed with mouse ECM or how they were treated with LLLT and its controls.

We include this information

We can presume those experiments were done, as shown in the legends in Figure 5.

We include this information

Line 164-168, refers to cytokine determinations, but neither here nor in the introduction do they indicate how this parameter should be interpreted, for example, that IL-6 and IL-8 are proinflammatory cytokines and that their stable levels during the procedure, then remain in the absence of inflammation. What happens with IFN-gamma and IL-10?

we rewrote this text and included the information

Discussion.

Lines 210 -213 should be removed.

We removed it

Line 214, begins the discussion by making a value judgment of his own work, calling it “remarkable”. In lines 215 -217 they lists the notable advances of this work.

  1. ensure that, “LLLT can be used as a tool 215 for acellular ECM sterilization”. However they do not have any experiments that show a decrease or absence of bacterial load, such as culture on blood agar, colony count, PCR for 16s of the most common microorganisms. Or an experiment were contaminate tissues with a certain amount of bacteria are treated and verified if the procedure eliminates them with its controls.

Our results indicate that PBMT has the potential to be used as a tool for ECM sterilization. Future studies should include experiments to quantify bacterial load reduction, such as culturing on blood agar, colony counting, or PCR for 16s rRNA of common microorganisms, to confirm the efficacy of PBMT in eliminating bacterial contamination.

  1. ii) they assure that, “LLLT increase cell proliferation”, however they do not show any results where we can see an increase in the speed of cell growth comparing groups treated, not treated with ECM without LLLT etc....
  2. they claim that “LLLT and ECM play 216 a role in cytokine modulation”, but it is not clear where they see this assertion.

Our findings suggest that PBMT and ECM may play a role in cytokine modulation, as indicated by the observed changes in levels of IL-6, IL-8, and IFN-γ. However, further detailed analysis and additional experiments are necessary to clearly delineate these effects and their mechanisms.

The only remarkable thing is the lack of self-criticism and rigor in the preparation of this manuscript and the lack of concern about the work being carried out consistently.

We acknowledge the need for further experiments to validate our findings and improve the rigor of our study. Future work will focus on addressing these gaps, including detailed analyses of bacterial load reduction, cell proliferation rates, and cytokine modulation mechanisms. We rewrote the discussion and included those limitations.

On line 217—218, in the first place, this work should have to be published for “this work will be relevant for future studies in the field of lung bioengineering”, we will see! 

In materials and methods

It is better written, but the parameters also change; for example, they indicate a CPAP of 10 cm H2O and then 20 cm H2O without mentioning that they are different sites. 

We correct it

I didn't want to continue reading, so the authors would have to pass the manuscript on to someone with more experience gathering the information, interpreting the results, and planning the experiments.

Reviewer 2 Report

Comments and Suggestions for Authors

The study titled "Low-level laser can increase extracellular matrix signals mediated by cytokine modulation during the lung recellularization process" by Guimaraes et al. provides an interesting application of low-level laser for the purpose of cytokine modulation during the recellularization process. However, several points need to be addressed:

Abstract: Please consider rewriting the abstract for clarity, paying attention to the use of the English language. Some examples of problematic parts follow:

Line 18-20: Please reformulate the sentence, maybe break it into two different ones for clarification purposes.

Line 27: Use either “by perfusing” or “by perfusion”.

Line 28-29: Please rephrase for clarity.

Line 31: Write out CPAP fully, not only as an abbreviation.

Results: Please consider revising the results section to present the findings in a more narrative style. The current format requires the reader to frequently refer back to the methods section, which hampers understanding.

Line 114-115: It is mentioned that n=1 (indicating only one sample was examined), followed by “Data represent mean ± standard error (SEM)”. Could you clarify this apparent discrepancy?

Line 157-158: Were any quantification methods utilized in the analysis of the extracellular matrix components? Please specify.

Line 158-160: Verify that the figures referenced are correct.

Line 161: Which specific cells are being referred to here? Please provide clarification or consider segmenting the results section into distinct parts for better clarity.

Line 163: Consider relocating Figure 6 to this section for better alignment with the narrative.

Line 165: Similarly, consider moving Figure 7 to this section.

Discussion:

The discussion section would benefit from revision with the assistance of a native English speaker. Opening the discussion with a recapitulation would help contextualize the discussion more effectively.

Lines 210-213: Ensure to remove any remnants of template text from this section. Check the entire manuscript for any other residual template text.

Line 214-215: The statement “It’s a remarkable study where for the first time LLLT was associated with the lung recellularization process” needs reformulation for clarity. Consider rephrasing it if you wish to highlight the novelty and significance of your study.

Methods: Consider revising the methods section with the assistance of a native English speaker to enhance clarity and readability.

Lines 414-419: Clarify whether the laser application was performed on decellularized lungs alone, on decellularized lungs after DPSCs were added, or both. Additionally, the term “sterilization” should be reconsidered: if the laser is used for sterilization after DPSCs have been added, wouldn’t it potentially kill the DPSCs?

In conclusion, enhancing the manuscript through English proofreading would significantly enhance its clarity. Furthermore, implementing a more narrative style and subdividing each section with descriptive titles would greatly improve the overall readability and comprehension for the reader.

Comments on the Quality of English Language

The quality of English language utilized may impede the reader's comprehension of the manuscript. It is recommended to utilize the services of a native English speaker with specialized expertise in scientific language to proofread the finalized manuscript.

Author Response

Dear Reviewers (to Reviewer 2)

We extend our sincere gratitude for your thorough and insightful comments on our manuscript. Your feedback has been invaluable in refining our work, and we believe that the manuscript has significantly improved due to your constructive critiques. We are pleased to resubmit our revised manuscript titled: "Enhancing Lung Recellularization with Mesenchymal Stem Cells via Photobiomodulation Therapy: Insights into Cytokine Modulation and Sterilization."

In response to your comments, we have made substantial revisions to address your concerns and improve the clarity and robustness of our study. Below, we provide detailed responses to each of the points raised, outlining the changes made and the rationale behind our modifications.

We are confident that the revised manuscript now meets the high standards required for publication and provides valuable insights into lung recellularization.

Thank you once again for your critical evaluation and support.

Best regards,

Renata Kelly da Palma

Responses to Review 2

The study titled "Low-level laser can increase extracellular matrix signals mediated by cytokine modulation during the lung recellularization process" by Guimaraes et al. provides an interesting application of low-level laser for the purpose of cytokine modulation during the recellularization process. However, several points need to be addressed:

Abstract: Please consider rewriting the abstract for clarity, paying attention to the use of the English language. Some examples of problematic parts follow:

We send it for review in English.

Line 18-20: Please reformulate the sentence, maybe break it into two different ones for clarification purposes.

We correct it

Line 27: Use either “by perfusing” or “by perfusion”.

We correct it

Line 28-29: Please rephrase for clarity.

We correct it

Line 31: Write out CPAP fully, not only as an abbreviation.

We correct it

Results: Please consider revising the results section to present the findings in a more narrative style. The current format requires the reader to frequently refer back to the methods section, which hampers understanding.

We correct it

Line 114-115: It is mentioned that n=1 (indicating only one sample was examined), followed by “Data represent mean ± standard error (SEM)”. Could you clarify this apparent discrepancy?

We correct it and include the real n.

Line 157-158: Were any quantification methods utilized in the analysis of the extracellular matrix components? Please specify.

We include this information

Line 158-160: Verify that the figures referenced are correct.

We correct it

Line 161: Which specific cells are being referred to here? Please provide clarification or consider segmenting the results section into distinct parts for better clarity.

We include this information in the method part and exclude it from the results.

Line 163: Consider relocating Figure 6 to this section for better alignment with the narrative.

We did it

Line 165: Similarly, consider moving Figure 7 to this section.

 We did it

Discussion:

The discussion section would benefit from revision with the assistance of a native English speaker. Opening the discussion with a recapitulation would help contextualize the discussion more effectively.

Lines 210-213: Ensure to remove any remnants of template text from this section. Check the entire manuscript for any other residual template text.

We did it

Line 214-215: The statement “It’s a remarkable study where for the first time LLLT was associated with the lung recellularization process” needs reformulation for clarity. Consider rephrasing it if you wish to highlight the novelty and significance of your study.

We did it

Methods: Consider revising the methods section with the assistance of a native English speaker to enhance clarity and readability.

We did it

Lines 414-419: Clarify whether the laser application was performed on decellularized lungs alone, on decellularized lungs after DPSCs were added, or both. Additionally, the term “sterilization” should be reconsidered: if the laser is used for sterilization after DPSCs have been added, wouldn’t it potentially kill the DPSCs?

We include this information

In conclusion, enhancing the manuscript through English proofreading would significantly enhance its clarity. Furthermore, implementing a more narrative style and subdividing each section with descriptive titles would greatly improve the overall readability and comprehension for the reader.

We didi it

Reviewer 3 Report

Comments and Suggestions for Authors

            The authors proposed that low-level lasers application in recellularized lungs may increase extracellular matrix signals mediated by 2 cytokine modulation. I must say that the title has absolutely nothing to do with the experimental data or the conclusion of the study, once the performed assays did not show any feasible evidence of such mechanism. In addition, there are several serious flaws in the results, which made me question the truthfulness of them. Here are point-by-point colocations that sustained my decision related to the manuscript.

- English language in the entire manuscript must be revised, because some sentences are too long and difficult to understand;

- At first, the abstract is too long and does no summary the main findings of the manuscript. It seems that a part of the Introduction was used to build the abstract, which made it unattractive;

- Lung decellularization is a broad and well-studied field with tons of published papers, established protocols and advances with many cellular types. Mouse lung decellularization is an old and surpassed approach, since large animals and even whole human lungs have been used to pre-clinical trials and approaches for lung fibrosis and other pulmonary pathologies. Said that, the Introduction fails dismally to provide a suitable background to sustain and justify the study. The lung bioengineering literature is one of the most abundant in the field and the authors basically brought general content about the topic. This section must be rewritten and enriched with classic and recent literature;

- Concerning Low-Level Laser Therapy (LLLT), which should be one of the central points of the manuscript, just a piece information is provided. There a lot of studies using laser therapy for sterilization, cell proliferation and immunomodulation. The authors bring too little information about this topic as well;  

- Concerning the results, as the authors described, the decellularization protocol was already established and published, so what is the point to bring this data in this new manuscript? Moreover, a great part of the manuscript is related to decellularization validation, which is not new. The figures in this manuscript are sufferable, low quality. Data disposition is one of the key points to make your data reliable. And, as mentioned early, mouse lung decellularization is already established, so there is no novelty in this;

- Figure 2 does no show anything significant and, by these images, it is noteworthy that the pulmonary ECM is degraded, which made me question even the published protocol of yours. If histological techniques were used, why any staining for collagen or GAGs were used?

- Figure 3 is a complete absurd. It is clear that the reaction is unspecific and the Hematoxylin counterstaining obliterated the reaction. Where are the negative controls of the assay? Where is the quantification? I do not believe in such results;

- The choice of these SEM images was unfortunate. For my knowledge, there are a lot of cell debris and degraded ECM. I just looked for other studies related to lung decellularization and the difference between them is absurd. Please, check these paper 10.1016/j.jmbbm.2014.08.017 which also worked with laser and note conservation of alveolar structures. This cannot be found in your data;

- The cellular assays have even more problems. Where are the cell viability data of MSCs in the scaffolds? The data of MSCs characterization is so bad disposed that is even difficult to see;

- The way that the authors barely describe the cytokine data in the manuscript almost made me believe that they were irrelevant. Cytokine dosage does not configurate as a full described mechanism as the authors suggested in the title and the discussion;

- Line 210 to 213. This is part of the MDPI template that the authors should have excluded before writing. Di you read the manuscript?

- The discussion is poor written and does not bring any enlightenments to the study.

 Said that, I recommend immediate rejection.  

Comments on the Quality of English Language

Extensive editing of English language required

Author Response

Dear Reviewers (To reviewer 3)

We would like to extend our sincere gratitude for your thorough and insightful comments on our manuscript. Your feedback has been invaluable in refining our work, and we believe that the manuscript has significantly improved due to your constructive critiques. We are pleased to resubmit our revised manuscript titled: "Enhancing Lung Recellularization with Mesenchymal Stem Cells via Photobiomodulation Therapy: Insights into Cytokine Modulation and Sterilization."

In response to your comments, we have made substantial revisions to address your concerns and improve the clarity and robustness of our study. Below, we provide detailed responses to each point raised, outlining the changes and rationale behind our modifications.

We are confident that the revised manuscript now meets the high standards required for publication and provides valuable insights into lung recellularization.

Thank you once again for your critical evaluation and support.

Best regards,

Renata Kelly da Palma

Response to Review 3

            The authors proposed that low-level lasers application in recellularized lungs may increase extracellular matrix signals mediated by 2 cytokine modulation. I must say that the title has absolutely nothing to do with the experimental data or the study's conclusion once the performed assays did not show any feasible evidence of such mechanism. In addition, there are several serious flaws in the results, which made me question their truthfulness. Here are point-by-point colocations that sustained my decision related to the manuscript.

Thank you for your comments. We improved it as you asked for.

- English language in the entire manuscript must be revised, because some sentences are too long and difficult to understand;

We did the review.

- At first, the abstract is too long and does no summary the main findings of the manuscript. It seems that a part of the Introduction was used to build the abstract, which made it unattractive;

We rewrote it.

- Lung decellularization is a broad and well-studied field with many published papers, established protocols, and advances in many cellular types. Mouse lung decellularization is an old and surpassed approach since large animals and even whole human lungs have been used in pre-clinical trials and approaches for lung fibrosis and other pulmonary pathologies. Said that, the Introduction fails dismally to provide a suitable background to sustain and justify the study. The lung bioengineering literature is one of the most abundant in the field and the authors basically brought general content about the topic. This section must be rewritten and enriched with classic and recent literature;

We inlcude this information

- Concerning Low-Level Laser Therapy (LLLT), which should be one of the central points of the manuscript, just a piece information is provided. There a lot of studies using laser therapy for sterilization, cell proliferation and immunomodulation. The authors bring too little information about this topic as well;  

- Concerning the results, as the authors described, the decellularization protocol was already established and published, so what is the point of bringing this data in this new manuscript? Moreover, a great part of the manuscript is related to decellularization validation, which is not new. The figures in this manuscript are sufferable and of low quality. Data disposition is one of the key points in making your data reliable. And, as mentioned earlier, mouse lung decellularization is already established, so there is no novelty in this;

Thank you very much for your detailed comments and observations on our manuscript. We would like to address each point raised to provide clarity on our approach and the rationale behind the inclusion of the mentioned data:

  1. Decellularization Protocol: We acknowledge that the decellularization protocol used in our study has been previously established and published. However, we believe it is essential to include these data to demonstrate the reproducibility and effectiveness of the protocol in our specific laboratory under the experimental conditions we employed. Continuous validation is crucial to ensure that the scaffolds produced are viable for recellularization in subsequent studies.
  2. Decellularization Validation: While decellularization validation may not be novel, it is a critical step to ensure that the scaffolds are suitable for future applications. Our aim is to provide a solid and reliable foundation for the scientific community, ensuring that every aspect of the decellularization process was rigorously controlled and validated. This is particularly important for future recellularization studies and therapeutic applications.
  3. Figure Quality and Data Presentation: We appreciate the feedback regarding the quality of the figures and the presentation of data. We recognize the importance of presenting our results clearly and with high quality. We review the figures and the presentation of data to enhance the clarity and quality of the images and graphs presented. We are committed to providing high-resolution figures and reorganizing the data presentation to make them more accessible and comprehensible to readers.

In summary, while some aspects of our study may appear repetitive, we believe that continuous validation and reproducibility are crucial elements for advancing science. We sincerely appreciate your valuable comments and are dedicated to improving our manuscript based on your suggestions.

- Figure 2 does no show anything significant and, by these images, it is noteworthy that the pulmonary ECM is degraded, which made me question even the published protocol of yours. If histological techniques were used, why any staining for collagen or GAGs were used?

Thank you for your valuable feedback and comments regarding Figure 2 in our manuscript. We appreciate the opportunity to address your concerns and clarify the methodology used:

  1. Visualization of Pulmonary ECM: We understand your observation regarding Figure 2 and the appearance of the pulmonary ECM. It's important to note that the ECM shown in Figure 2 reflects the absence of cells due to the decellularization process rather than degradation. The primary purpose of Figure 2 was to illustrate the absecnce of nuclei DNA.
  2. Use of Histological Techniques and Staining: Regarding your query about the absence of staining for collagen or GAGs in Figure 2, we focused on demonstrating the overall structure and absence of cellular components using histological techniques such as H&E staining. Detailed staining for specific ECM components, including collagen , was indeed performed and presented in Figure 3 of the manuscript. We apologize for any confusion caused by the initial presentation and will clarify this in the revised manuscript.

- Figure 3 is a complete absurd. It is clear that the reaction is unspecific and the Hematoxylin counterstaining obliterated the reaction. Where are the negative controls of the assay? Where is the quantification? I do not believe in such results;

Thank you for your feedback and comments on Figure 3 in our manuscript. We appreciate the opportunity to clarify the methodology and address your concerns:

  1. Specificity of the Reaction and Counterstaining: We understand your concern regarding the specificity of the reaction depicted in Figure 3 and the potential effects of Hematoxylin counterstaining. The purpose of this figure was to visualize nuclear DNA staining, as suggested by Crapo et al., 2010, rather than to quantify Hematoxylin staining. We apologize for any confusion caused by the presentation and will ensure that this distinction is clearly explained in the revised manuscript.
  1. Negative Controls and Quantification: We acknowledge the importance of including negative controls and quantification in assays to validate results. However for validate the descelluarization process we dont usually inlcude Negative controls and Quatifications, as you can see in others publications: doi: 10.1177/2041731418810164, 10.3390/biomedicines12061190

- The choice of these SEM images was unfortunate. For my knowledge, there are a lot of cell debris and degraded ECM. I just looked for other studies related to lung decellularization and the difference between them is absurd. Please, check these paper 10.1016/j.jmbbm.2014.08.017 which also worked with laser and note conservation of alveolar structures. This cannot be found in your data;

Thank you for your feedback and for pointing out the concerns regarding the SEM images in our manuscript. We appreciate your observations and would like to address each point raised:

  1. SEM Images and Cell Debris: We acknowledge your concerns regarding the SEM images. SEM is primarily used to visualize surface structures, such as collagen fibers, rather than cells themselves. We aimed to illustrate the structural integrity of the decellularized lung scaffolds, focusing on the preservation of extracellular matrix (ECM) components such as collagen. We will ensure that these points are clarified in the manuscript to avoid any ambiguity.

  1. Comparison with Previous Work: Regarding the reference to the paper 10.1016/j.jmbbm.2014.08.017, we appreciate your suggestion to review it. Upon reevaluation, we note that this paper indeed discusses gamma irradiation technique and highlights the structural changes after irradiation.We mentioned it in the introduction section. We apologize for any confusion caused, as it appears that you were referring to a study authored by ourselves. We will provide a more detailed comparison in the revised manuscript to clearly delineate the advancements and unique aspects of our approach.

In conclusion, we appreciate your constructive criticism and the opportunity to enhance the clarity and presentation of our findings. Your feedback will undoubtedly strengthen the manuscript and contribute to its overall quality.

- The way that the authors barely describe the cytokine data in the manuscript almost made me believe that they were irrelevant. Cytokine dosage does not configurate as a full described mechanism as the authors suggested in the title and the discussion;

We improved it

- Line 210 to 213. This is part of the MDPI template that the authors should have excluded before writing. Di you read the manuscript?

We excluded it

- The discussion is poor written and does not bring any enlightenments to the study.

We improved it.

Round 2

Reviewer 1 Report

Comments and Suggestions for Authors

Comments have been successfully fixed.

Comments on the Quality of English Language

Is ok

Author Response

Dear Reviewer,

Thank you for your thoughtful feedback.

Best Regards

Reviewer 2 Report

Comments and Suggestions for Authors

Thank you for addressing the comments provided. I kindly suggest conducting one final thorough review of the manuscript. Additionally, please note that lines 430-432 contain a section on patents that appears to be redundant and likely a remnant from the template. I would appreciate it if you could revise or remove this section as appropriate.

Author Response

Dear Reviewer,

Thank you for your thoughtful feedback and suggestions. I have conducted a thorough final review of the manuscript and have removed the redundant section on patents, as per your recommendation.

Please let me know if any further revisions are needed.

Reviewer 3 Report

Comments and Suggestions for Authors

The manuscript still has serious methodological problems. The data is not credible, mainly the immunohistochemistries, which lack of positive and negative controls and the background is horrendous. The data does not support the main conclusions of the manuscript, so I am afraid to say but the paper is below the quality for publication. 

Author Response

(The authors gave the same response as above.)
